# Impact of Disease-Specific Fears on Pulmonary Rehabilitation Trajectories in Patients with COPD

**DOI:** 10.3390/jcm8091460

**Published:** 2019-09-13

**Authors:** Thomas Janssens, Zora Van de Moortel, Wolfgang Geidl, Johannes Carl, Klaus Pfeifer, Nicola Lehbert, Michael Wittmann, Konrad Schultz, Andreas von Leupoldt

**Affiliations:** 1Health Psychology, KU Leuven (University of Leuven), 2800 Leuven, Belgium; 2Department of Sport Science and Sport, Friedrich-Alexander University Erlangen-Nürnberg, 91058 Erlangen, Germany; 3Clinic Bad Reichenhall, Centre for Rehabilitation, Pneumology and Orthopedics, 83435 Bad Reichenhall, Germany

**Keywords:** disease-specific fears, COPD, pulmonary rehabilitation, anxiety

## Abstract

Disease-specific fears predict health status in chronic obstructive pulmonary disease (COPD), but their role in pulmonary rehabilitation (PR) remains poorly understood and especially longer-term evaluations are lacking. We therefore investigated changes in disease-specific fears over the course of PR and six months after PR, and investigated associations with PR outcomes (COPD assessment test (CAT) and St. Georges respiratory questionnaire (SGRQ)) in a subset of patients with COPD (*n* = 146) undergoing a 3-week inpatient PR program as part of the STAR study (Clinicaltrials.gov, ID: NCT02966561). Disease-specific fears as measured with the COPD anxiety questionnaire improved after PR. For fear of dyspnea, fear of physical activity and fear of disease progression, improvements remained significant at six-month follow-up. Patients with higher disease-specific fears at baseline showed elevated symptom burden (CAT and SGRQ Symptom scores), which persisted after PR and at follow-up. Elevated disease-specific fears also resulted in reduced improvements in Quality of Life (SGRQ activity and impact scales) after PR and at follow-up. Finally, improvement in disease-specific fears was associated with improvement in symptom burden and quality of life. Adjustment for potential confounding variables (sex, smoking status, age, lung function, and depressive symptoms) resulted in comparable effects. These findings show the role of disease-specific fears in patients with COPD during PR and highlight the need to target disease-specific fears to further improve the effects of PR.

## 1. Introduction

Chronic obstructive pulmonary disease (COPD) is a prevalent respiratory condition characterized by airflow obstruction, mainly due to inflammatory lung pathology caused by exposure to cigarette smoke or other noxious particles and gases [1]. Common symptoms of COPD include dyspnea, cough, sputum production, wheezing, and chest tightness, which are associated with a significant individual, social, and economic burden [1].

Pulmonary rehabilitation (PR) is an effective and evidence-based intervention for COPD [1]. PR is a multidisciplinary treatment strategy, which consists of exercise training, respiratory physiotherapy, smoking cessation, and patient education as key components, with the goal of behavior change realized through encouraging self-efficacy [2]. PR has been shown to improve exercise capacity, dyspnea, emotional functioning, as well as health-related quality of life [3,4]. Nevertheless, key questions remain about the essential components and mediating mechanisms of PR as well as on the long-term efficacy of PR in accomplishing and maintaining behavior change [4,5].

Anxiety and depression are highly prevalent in patients with COPD, with prevalence estimates ranging from 10% to 50% [6,7,8,9,10,11]. Furthermore, these psychological disorders are often underdiagnosed and undertreated in patients with COPD [8,12,13]. High levels of anxiety and depression are associated with reduced quality of life [14], increased costs to the healthcare system [15,16], and increased mortality [17,18]. Furthermore, anxiety and depression are associated with reduced functional exercise capacity and increased dyspnea [19,20], increased hospitalizations and COPD exacerbations [21,22,23,24], as well as unfavorable health behaviors including reduced self-reported medication adherence [25], and increased levels of smoking and physical inactivity [26].

Most previous studies on anxiety and COPD have only examined general forms of anxiety, irrespective of its link to COPD. However, researchers have started to explore the role of disease-specific fears and anxiety in COPD [27,28], which may be more closely related to relevant symptoms and behaviors in COPD, and therefore provide a target for more specific interventions [29,30]. In the field of chronic pain, this approach has led to a greater understanding of the sensation of pain and pain-related disability, and has spurred interventions aimed at reducing pain-related fear [31,32]. Recent findings in COPD show that worse dyspnea-related anxiety and fear of physical activity are associated with indicators of symptom severity, quality of life, exercise capacity, and even structural brain changes [29,30,33,34,35,36,37,38]. Other fears identified in patients with COPD include fear of social exclusion, fear of disease progression and sleep-related worries [33]. All five categories have shown associations with disease-specific disability, and provide incremental predictive value above measures of general negative affect (anxiety and depression) [30].

Disease-specific fears may help or hinder currently available treatments for COPD. Qualitative studies have suggested that disease-specific fears interfere with self-management of COPD and are associated with concerns about participation in pulmonary rehabilitation [28,39]. Furthermore, several studies have reported that symptom-specific fears can improve during PR [37,40,41,42,43,44], whereas there is suggestive evidence that symptom-specific fears remain stable in usual care [40]. These findings would fit with an interpretation that repeated exposure to physical activity during PR would reduce disease-specific fears, especially fear of dyspnea and fear of physical activity. However, knowledge on the maintenance of these changes after rehabilitation is currently lacking, but would offer insight into the role of disease-specific fears in maintaining improved PR outcomes.

A related question is whether disease-specific fears at the start of PR help or hinder the outcome of rehabilitation. There are some initial findings that individuals starting PR with higher levels of fear of dyspnea and fear of physical activity show greater improvements in exercise capacity during PR, as well as a greater reduction in exercise-induced dyspnea and emotional burden [36]. Furthermore, participants characterized by greater activity limitations and fear of dyspnea but less severe airway obstruction show largest improvements after PR [40]. However, other contrasting findings suggest that disease-specific fears at baseline show persistent negative associations with PR outcomes including health status, quality of life, and 6 min walking distance, even after participation in PR [37,43,44,45], but investigations into associations of disease-specific fears with longer-term outcomes are still lacking.

With this study, we therefore aimed to investigate longer-term changes in disease-specific fears in PR participants with COPD as well as associations of disease-specific fears with PR outcomes. In accordance with previous findings, we expected a reduction in disease-specific fears over the course of a three-week inpatient PR. Furthermore, we expected these reductions to be maintained at six-month follow-up. Finally, we investigated the effects of baseline levels of disease-specific fears on PR outcome trajectories as well as the association of within-individual change in disease-specific fears on changes in PR outcomes, with symptom burden measured using CAT designated as the main outcome variable.

## 2. Experimental Section

### 2.1. Participants

Participants were the initial 146 participants who had completed participation in a larger, controlled trial of pedometer based behavioral intervention as an add-on to PR for COPD (STAR Study, Trial registration Clinicaltrials.gov, ID: NCT02966561. [46]). Data from these 146 participants consisted a first data release from the STAR Study investigators, with data blinded for group allocation. The study protocol was approved by the independent Research Ethics Committee of the Medical Faculty of Friedrich-Alexander-Universität Erlangen-Nürnberg, Germany (Ref. No. 321_15B). The main inclusion criterion was an ICD-10 diagnosis of COPD, confirmed by lung function (FEV1/FVC < 0.7 after bronchodilatory medication). Main exclusion criteria were severe comorbidity (e.g., cancer or severe cardiac or neurological comorbidities, severe psychiatric condition), considerable reduction of sight or hearing, or an inability to speak German [46].

### 2.2. Pulmonary Rehabilitation Program

Patients completed a three-week inpatient PR program. The PR program consists of obligatory elements (medication review, structured COPD patient education with practical training of correct inhalation technique (“device training”), endurance and strength exercise, whole body vibration training, and respiratory physiotherapy), with a main focus on exercise training. Additional optional elements included a comprehensive smoking cessation program, psychological interventions and social counseling, nutritional counseling, mucolytic physiotherapy, inspiratory muscle training, saline inhalation therapy, patient education regarding long-term oxygen use, and occupational therapy. Participants were randomized to receive either information on physical activity and physical activity exercises (control group), or a pedometer assisted intervention focused on instruction (how to perform monitoring and physical activity), monitoring, goal setting, and feedback (intervention group). For full details of the program, including a list of behavior change techniques employed by both treatment arms, see [46].

### 2.3. Measurements

#### 2.3.1. Lung Function and Exercise Capacity

Spirometry was performed in accordance with ERS/ATS guidelines [47], using reference values from the Global Lung Initiative [48]. Forced Expiratory Volume in 1 s (FEV1) and Vital Capacity (VC) were extracted as variables of interest.

The Six-Minute Walking Test (6MWT) was performed to evaluate the functional exercise capacity of the participants. The test was carried out on a 30 m track, in accordance with ATS guidelines [49]. However, a single test was performed as research in the context of a previous trial showed no added benefit of duplicate testing [50]. After 6 min, walking distance of participants was recorded. The test has appropriate reliability and validity for evaluation of exercise capacity in the context of PR [51].

#### 2.3.2. Self-Report Measures

The validated COPD Angst Fragebogen (CAF; COPD Anxiety Questionnaire, [33] was used to measure the disease-specific fears in patients with COPD. The CAF is a 27-item questionnaire, consisting of five subscales: fear of social exclusion (FSE) (eight items), fear of dyspnea (FD) (five items), fear of physical activity (FPA) (six items), fear of disease progression (FP) (six items), and sleep related worries (SRW) (two items). Each item is scored on a five-point Likert scale, from 0 (never) to 4 (always), with higher scores indicating higher levels of disease-specific fears. Two additional subscales related to long-term oxygen therapy and to partnership were assessed but not included in analyses, because these were not applicable for many patients. So far, no analysis of minimum clinically important differences (MCIDs) have been conducted for the CAF. Therefore, standardized mean differences (Cohen’s d) are presented below when discussing changes in CAF scores.

The COPD assessment test (CAT, [52]) was used as a measure of health status. The CAT consists of eight items assessing symptoms, activity limitations, and impact of COPD with answer scores ranging from 0 (no impairment) to 5 (maximal impairment). It is a brief instrument with good reliability and validity [53,54]. The MCID for the CAT ranges from 2 (anchor based) to 3.8 (distribution based) [54], with a pooled estimate of 3.29 in a German PR population [55].

The impact of dyspnea on daily activities in patients with COPD was evaluated using the German self-administered version of the baseline dyspnea index (BDI) and the transition dyspnea index (TDI) [56,57,58]. Both the BDI and the TDI are composed of three domains: functional impairment, magnitude of task, and magnitude of effort. The BDI measures the perceived severity at a single point in time and is the initial baseline assessment. The score on the BDI ranges from 0 (most severe level) to 12 (least physical limitation). The TDI measures change in comparison to the baseline assessment. Scores on the TDI range from −9 (more deterioration) to +9 (amelioration). The German versions of the BDI and the TDI maintain their concurrent and construct validity and can be used to evaluated changes in activity-related dyspnea in patients with COPD [56,57].

The St. Georges respiratory questionnaire (SGRQ, [59]) was used to measure the health-related quality of life (HRQoL) in patients with COPD. The SGRQ is a valid and reliable instrument to measure the self-reported health impairment and HRQoL in COPD patients [53]. The SGRQ consists of three subscales. The impact subscale measures social functioning and psychological disturbances resulting from airway disease; the symptoms subscale measures the effect of respiratory symptoms, their frequency and severity; and the activity subscale looks at activities that cause or are limited by dyspnea. Scores represent a percentage of overall impairment where 0 represents the best possible HRQoL and 100 the worst possible HRQoL.

Depressive symptoms were measured using the Patient Health Questionnaire depression scale (PHQ-9) [60]. This questionnaire consists of nine questions; each scored on a scale ranging from 0 (not at all) to 3 (nearly every day). The total score ranges from 0 to 27, with higher scores indicating more depressive symptoms. The German version of the PHQ is found to be a reliable and valid measure for depression in patients with COPD [11].

### 2.4. Procedure

Patients filled out all self-report measures (CAF, CAT, BDI/TDI, SGRQ, and PHQ-9) at the start, at the end, and six months after the pulmonary rehabilitation program. FEV1 and 6MWT distance were only available at the start and end of the pulmonary rehabilitation program. For a full overview of measures available in the dataset at each measurement time, see [46].

### 2.5. Data Analysis

Changes in patients’ outcomes across the three measurement points were analyzed using linear mixed models. In order to model change over time, we used fixed effects of timepoint (three levels—start of PR, end of PR, six-month follow-up), as well as random intercept and slopes, implemented using an unstructured variance/covariance matrix. We carried out separate analyses for the different outcome measures (BDI, TDI, 6MWT, SGRQ, and CAT). The control variables in our analysis were gender, age, smoking status, lung function (as measured by the FEV1% predicted), and depression (as measured by the PHQ-9).

Impact of disease-specific fears on outcome trajectories were analyzed in a first set of models using baseline (continuous) CAF scores and their interaction with time as a fixed predictor variable. In order to facilitate interpretation of these continuous predictors, baseline CAF scores were standardized and comparisons of participants scoring −1, 0, and 1 (corresponding to CAF scores 1SD below mean, at mean, and 1SD above mean) are reported. In a second set of analyses, we included (continuous) CAF scores as a time-varying covariate. In these analyses, CAF scores were mean centered within participants in order to assess effects of within-individual change in CAF scores. Scores were standardized using baseline SD for each disease-specific fear in order to facilitate comparisons. These analyses also included mean CAF for each participant (standardized across participants) to account for effects of between-individual differences in CAF scores. Models were fitted separately for each CAF subscale and each outcome variable; and corrected analyses also included additional control variables: gender, age, smoking status, baseline lung function (FEV1% predicted), and baseline depressive symptoms (PHQ-9). For effect plots, we opted to not only include estimated trajectories (output of the different models), but also plotted observed data in order to provide a visual check for model fit and underlying variability in the data.

We performed all analyses using R (lme4, lmerTest, and emmeans packages) [61,62,63], using Satterthwaite’s approximation for degrees of freedom and *p*-values. The significance level was set at *p* ≤ 0.05 (using Tukey corrections for multiple comparisons).

## 3. Results

### 3.1. Patient Characteristics

We studied an initial sample of 146 patients who participated in the STAR Study. These data consisted a first, blinded data release from the STAR Study investigators, and no data on group allocation was available. Patients had an average age of 57.54 years (SD = 4.38). Amongst the patients were 45.2% active smokers and 54.8 nonsmokers with a mean of 44.39 pack years (SD = 22.80). A summary of other patient characteristics can be found in Table 1. Age, gender, and inhaled medication use were not associated with disease-specific fears at baseline. Exacerbation history, FEV1, 6 min walking distance, and depressive symptoms showed associations with three or more disease-specific fears.

### 3.2. Change in Disease-Specific Fears and other Pulmonary Rehabilitation Outcomes

Patients showed significant changes on all subscales of the COPD anxiety questionnaire across time (Table 2). Further exploration showed a significant improvement for all disease-specific fears at the end of PR (Cohen’s *d* 0.48 (fear of social exclusion)–1.26 (fear of dyspnea), Figure 1). For fear of dyspnea (*d =* 1.24, *p* < 0.0001), fear of physical activity (*d =* 0.75, *p* < 0.0001), and fear of disease progression (*d =* 0.52, *p =* 0.0004), these improvements remained significant at 6 months follow-up, whereas this was not the case for fear of social exclusion (*d =* 0.18, *p =* 0.3340) and sleep related worries (*d =* 0.31, *p =* 0.0981), (Figure 1).

When controlling for covariates, the main effects of time remained significant for all CAF subscales, and effect sizes were similar to the primary analyses: Start–End PR: *d =* 0.48 (fear of social exclusion)–*d =* 1.26 (fear of dyspnea); Start–Follow Up: *d =* 0.18 (fear of social exclusion)–*d =* 1.25 (fear of dyspnea); see Appendix A for full results.

At the end of pulmonary rehabilitation, participants had improved on all outcomes. For our main outcome (Symptom burden measured using CAT), participants showed an average improvement from Start–End PR of 5.54 points (SE = 0.475, *d =* 1.412, *p* < 0.0001). At six months follow up, improvement was reduced to 2.45 points (SE = 0.513, *d =* 0.625, *p* < 0.0001). Similar outcomes were found for reductions in SGRQ subscales (Start–End PR: *d =* 0.733 (SGRQ Activity)–*d =* 1.144 (SGRQ Impact); Start–Follow Up: *d =* 0.449 (SGRQ Activity)–*d =* 0.812 (SGRQ Impact)). Six-minute walking distance was improved by 81.5 m at the end of PR (SE = 9.06, *d =* 1.19, *p* < 0.0001). Furthermore, participants reported a change in impact of dyspnea on daily living (measured using TDI), but these changes were reduced at six months follow up. Improvements in depressive symptoms (Start–End PR: 2.87, SE = 0.382, *d =* 0.919, *p* < 0.0001), were no longer significantly different from baseline at follow up (0.797, SE = 0.398, *d =* 0.251, *p =* 0.1218), (Table 2). When taking control variables into account, findings were similar for all outcome measures (see Appendix A for full results).

### 3.3. Impact of Disease-Specific Fears on Pulmonary Rehabilitation Outcomes

#### 3.3.1. Baseline Levels of Disease-Specific Fears

Baseline levels of all disease-specific fears had a main effect on symptom burden. Fear of progression showed the smallest association with CAT scores; with baseline fear of progression 1SD above the mean being associated with 2.53 (1.47–3.58) points higher CAT scores. Sleep-related worries showed the largest associations; with baseline scores 1SD above the mean being associated with 3.76 (2.82–4.71) points higher CAT scores. Associations between baseline disease-specific fears and CAT scores were similar after PR and follow up (Figure 2). Disease-specific fears did not have different effects at different time points (all interaction terms *p* > 0.32). Main effects of disease-specific fears remained significant after inclusion of covariates, although their magnitudes were attenuated, ranging from 1.30 (0.25–2.36) for fear of progression to 2.81 (1.87–3.74) for sleep-related worries.

Disease-specific fears had similar main effects on SGRQ symptom scores, with higher baseline levels of disease-specific fears being associated with higher SGRQ symptom scores across all time points. Effects ranged from 5.87 (2.50–9.25) points for participants with baseline fear of progression 1SD above the mean to 10.39 (7.27–13.50) for sleep-related worries. After adjusting for covariates, the main effects of disease-specific fears remained significant, but the estimates were somewhat attenuated, ranging from 4.7 (1.27–8.12) points for fear of progression to 7.78 (4.62–10.9) for sleep-related worries.

Baseline differences in disease-specific fears were not associated with differences in SGRQ activity scores at baseline. However, differences in SGRQ activity scores emerged after PR and at follow-up, resulting in reduced treatment efficacy for participants with higher disease-specific fears (Figure 3). Differential treatment effects were similar for all disease-specific fears, with smallest effects for fear of dyspnea, and largest effect for fear of physical activity. Participants with average fear of dyspnea improved 11.09 (5.05–17.13) points during PR (*d =* 0.74, *p* < 0.0001), which was reduced to 2.89 (−4.66–12.43) points (*d =* 0.26, *p =* 0.8886) for participants having fear of dyspnea 1SD above the mean. Similarly, participants with mean levels of fear of physical activity improved 11.10 (5.15–17.06) points during PR (*d =* 0.76, *p* < 0.0001), whereas this was reduced in participants with fear of physical activity 1SD above the mean (1.89 (−6.52–10.3), *d =* 0.13, *p =* 0.9987). After PR, differences in SGRQ activity scores between participants with high vs. mean levels of disease-specific fears were smallest for sleep-related worries 6.64 (1.73–11.56) and largest for fear of physical activity 9.28 (4.53–14.03). At follow up, we observed smallest effects for fear of dyspnea 7.93 (1.54–14.31) and largest effects for fear of physical activity 9.91 (3.80–16.02). Adjusting for covariates, estimates were somewhat attenuated, but remained significant. After PR, this resulted in smallest effects for fear of progression 5.59 (0.29–10.88) and largest effects for fear of physical activity 8.10 (3.04–13.17). Similar results were obtained at follow-up (fear of progression 5.89 (0.89–12.67), fear of physical activity 8.67 (2.32–15.02).

Similar to the effects of disease-specific fears on SGRQ activity scores, baseline differences in disease-specific fears were not associated with differences in SGRQ impact scores at baseline, but emerged after PR and at follow-up, resulting in reduced treatment efficacy for participants with higher disease-specific fears. In these analyses, differences in SGRQ impact scores were similar for all disease-specific fears, with differences between participants with high vs. mean levels of disease-specific fears being smallest for fear of dyspnea (After PR: 7.43 (3.11–11.74), Follow Up: 6.62 (1.28–11.96)) and largest for sleep-related worries (After PR: 9.47 (5.34–13.60), Follow Up: 8.45 (3.36–13.55)). Adjusting for covariates, estimates where somewhat attenuated, but the effects remained significant with smallest effects observed for fear of dyspnea (After PR: 5.95 (1.52–10.37), Follow Up: 5.26 (0.09–10.61) and largest effects observed for sleep-related worries (After PR: 7.67 (3.30–12.03), Follow Up: 6.74 (1.52–11.96)).

#### 3.3.2. Associations between Changes in Disease-Specific Fears and Pulmonary Rehabilitation Outcomes

For all disease-specific fears, participant mean levels (across all time points) were strongly correlated with disease-specific fears at baseline (r range: 0.88–0.90), whereas baseline levels and within-individual changes (deviation from participant mean) showed smaller correlations (ranging from *r* = 0.27 (fear of social exclusion) to *r* = 0.52 (sleep-related worries) (see Appendix A for full correlation table). For all disease-specific fears, both participants mean levels and individual changes in disease-specific fears were independently associated with changes in CAT scores. Effects of individual changes were smallest for sleep-related worries, with a 1 SD reduction leading to a 1.66 (0.84–2.47) point reduction in CAT scores. Effects were largest for fear of physical activity, with a 1 SD reduction leading to a 2.57 (1.81–3.33) point reduction in CAT scores (Figure 4). Reductions in CAT ranged from 1.39 (0.58–2.20) (sleep-related worries) to 2.41 (1.66–3.17) (fear of physical activity) in analyses controlling for covariates. Across participants, effects for all disease-specific fears on CAT scores paralleled results of the analysis of baseline disease-specific fears, with smallest effects of fear of progression (3.40 (2.49–4.32), and largest effects of sleep-related worries (4.62 (3.87–5.37)) (Figure 4).

Effects were similar for SGRQ symptom scores, with within-participant reduction of 1 SD in all disease-specific fears leading to a reduction in SGRQ symptom scores ranging from 5.31 (2.55–8.10) (fear of physical activity) to 7.71 (5.09–10.48) (sleep-related worries). Reductions in SGRQ symptom scores ranged from 5.00 (1.91–8.08), (fear of physical activity) to 7.62 (4.77–10.47) (sleep-related worries) in analyses controlling for covariates. Across participants, 1 SD higher disease-specific fear scores were associated with higher SGRQ symptom scores, ranging from 9.52 (6.74–12.36) for social exclusion), to 12.23 (9.69–14.73), for sleep-related worries.

For SGRQ activity scores, within-participant reductions in fear of physical activity (5.42 (2.35–8.94)), fear of progression (4.97 (1.26–8.34)), and fear of social exclusion (3.88 (0.36–7.17)) were associated with improved outcomes. However, improvements in fear of dyspnea (0.81 (−3.04–4.47)) and sleep-related worries (2.26 (−1.06–5.85)) did not show significant associations with SGRQ activity scores. Analyses controlling for covariates showed similar results (fear of physical activity 5.80(2.40–9.59), fear of progression 5.34 (1.84–9.25), fear of social exclusion 4.60 (0.75–8.02), fear of dyspnea 0.94 (−3.31–5.42), sleep-related worries 2.91 (−0.80–6.33)). Across participants, all disease-specific fear scores were associated with higher SGRQ activity scores, ranging from 5.46 (3.28–7.50)) for fear of dyspnea to 8.35 (6.35–10.43) for fear of physical activity.

For SGRQ impact scores, within-participant reductions on all disease-specific fears were associated with reduced impact of COPD on daily life, with associations ranging from 3.40 (0.25–6.25), for changes in fear of dyspnea to 8.04 (5.38–10.71) for changes in fear of physical activity. Controlling for covariates yielded similar results, with associations ranging from 4.30 (1.00–7.53) for fear of dyspnea to 8.60 (5.93–11.39) for fear of physical activity. Across participants, all disease-specific fear scores were associated with higher SGRQ activity scores, ranging from 6.58 (4.58–8.57), for fear of dyspnea to 9.90 (8.27–11.53), for fear of physical activity.

## 4. Discussion

In this study, we investigated associations between disease-specific fears and changes in pulmonary rehabilitation outcomes during a three-week inpatient rehabilitation program and at six-month follow up. For all measurers of disease-specific fears, as well as all outcome measures, participants showed improvement during rehabilitation. Outcomes at six-month follow-up remained improved when compared to the start of pulmonary rehabilitation, although improvements in symptom burden and quality of life showed significant attenuation during follow-up compared to the end of rehabilitation. However, at six-month follow-up, depressive symptoms, fear of social exclusion, and sleep-related worries no longer showed improvement compared to the start of PR. Finally, across all timepoints, improvements in disease-specific fears were associated with better rehabilitation outcomes.

Notably, different disease-specific fears showed different change trajectories. We observed the largest reductions in fear of dyspnea and fear of physical activity. Other studies have reported changes in fear of dyspnea, fear of physical activity or related measures of fear and anxiety in the context of physical activity during pulmonary rehabilitation [37,43,64], and have reported correlations with changes in key outcomes of rehabilitation [37,42], but our study is the first to investigate longer-term changes in disease-specific fears. Additionally, other researchers have argued that changes in dyspnea-specific fear and anxiety may be a key mechanism for symptom change during pulmonary rehabilitation [42,65]. In our study, we showed that improvements in fear of physical activity were associated with improved symptom burden and quality of life. Effects of changes of fear of dyspnea were smaller, and failed to reach significance for the SGRQ activities subscale. The latter finding is unexpected, but could be a result of the specific content of the SGRQ activities subscale, which mainly focuses on physical activity. Nevertheless, in general our findings confirm previous findings on the importance of fear of dyspnea and fear of physical activity as important contributors to symptom burden and quality of life in individuals with COPD.

Changes in sleep-related worries and fear of social exclusion were smallest, and were no longer significantly different from baseline when measured at follow up. Poor sleep quality and sleep disorders are prevalent in patients with COPD, with over half of individuals with COPD reporting poor sleep quality, which contributes to reduced quality of life [66,67,68]. Similarly, loneliness and feelings of social exclusion have been documented in individuals with COPD, with similar impact on quality of life [69]. However, these issues were not an explicit target of the multicomponent pulmonary rehabilitation program. Furthermore, a short inpatient program may have limited potential for addressing and changing these issues. For example, although the social/group aspect of pulmonary rehabilitation has been suggested as a beneficial treatment component [70], a change in social context and transition back to the home context could be related to increased feelings of social exclusion. Nevertheless, within-individual changes in sleep-related worries and fear of social exclusion were associated with improvements in symptom burden and quality of life, highlighting the role of these fears as a potential target for treatment in individuals with COPD.

Higher baseline levels of disease-specific fears were associated with worse symptom burden at all time points, suggesting that the negative impact of disease-specific fears on symptoms persists even when COPD symptoms are being treated. Furthermore, participants with higher baseline levels of disease-specific fears showed less beneficial effects of pulmonary rehabilitation on self-reported activity limitations (SGRQ activity and impact scales), despite an improvement in symptom burden. These findings suggest that disease-specific fears may be especially important for the transfer of improvements in the rehabilitation context to activities in daily life, which has been identified as an ongoing challenge for the efficacy of pulmonary rehabilitation [5].

Although our findings on the negative effects of baseline disease-specific fears corroborate previous findings [37], other studies have found positive or inconsistent effects of baseline disease-specific fears on pulmonary rehabilitation outcomes [36,40,43]. This suggests that some rehabilitation programs may be more successful than others in mitigating effects of disease-specific fears. Although pulmonary rehabilitation programs are similar in focusing on exercise and patient education, there are also differences in patient or program characteristics, as well as duration or inpatient vs. outpatient setting of programs, which could have an impact on disease-specific fears and complicate the potential for generalization of findings to other rehabilitation programs [71,72]. For example, studies that have reported associations between disease-specific fears and improved rehabilitation outcomes were carried out in outpatient settings [36,40,43], whereas the current study confirms associations with worse treatment outcomes, which were previously observed in inpatient PR programs [37]. Replication in different settings, or direct comparisons of the efficacy of different (add on) treatment components could inform the development of rehabilitation programs that have increased efficacy in reducing disease-specific fears. Moreover, treatment components that explicitly target disease-specific fears have been mostly lacking in pulmonary rehabilitation programs. Exceptions include an add-on panic prevention program, using cognitive techniques to change catastrophic interpretations of breathlessness [73] and a more comprehensive CBT program, which included a combination of cognitive techniques, and exposure to feared activities to reduce fear and anxiety [74]. Further integration of these techniques during pulmonary rehabilitation (especially during exercise training) would be an interesting novel approach to further reduce fear of dyspnea and fear of physical activity [75].

Differences in outcomes associated with baseline levels of disease-specific fears suggest that assessment of disease-specific fears could be used as a basis for additional tailoring of COPD treatment. Many rehabilitation programs already provide some tailoring, including tailoring based on psychological needs [5,76]. As specific tailoring or treatment approaches aimed at reducing disease specific fears are currently lacking in COPD, it can be beneficial to compare the efficacy of existing approaches in reducing disease-specific fears, as well as explore novel tailoring approaches incorporating disease-specific fears. In such an approach, disease-specific fears become a treatable trait, that can be assessed and for which treatment can be incorporated in current COPD treatments [77].

There are a number of limitations which may reduce the generalizability of our findings. A first set of limitations relates to the specific setting in which we conducted the research: a single center offering inpatient rehabilitation. As noted above, there may be limitations to studying disease-specific fears during inpatient rehabilitation, as this setting is very different from the home setting of patients. Furthermore, we conducted an observational study in the context of a clinical trial that was not explicitly aimed at changing disease-specific fears, which leaves room for alternative causal pathways. One such potential cause of differential outcomes may be the optional treatment components that were part of the PR program, as it is conceivable that disease-specific fears may have an impact on the treatment components that are offered to patients. Unfortunately, we did not have data on specific patient trajectories or treatments that were offered to individual patients. Taken together, these limitations highlight the complexity of the relationship between disease-specific fears and PR outcomes. Additional research on disease-specific fears in other treatment settings, or research focusing on the effects of specific treatment components may be needed to clear up some of this complexity. A further limitation is the use of a subsample of an ongoing clinical trial: care in clinical trials may differ from usual care, which may have an impact on the role of disease-specific fear on outcomes achieved in this setting. Furthermore, the use of a limited dataset (data release with 146 completers) and the lack of information on treatment allocation may have reduced power. A final limitation relates to the measurement of disease-specific fears. There is currently no gold standard measure to assess these fears, and information on the clinical significance of changes in disease-specific fears (i.e., minimal clinically important difference) is lacking, which at present limits the use of these measures for routine clinical purposes.

## 5. Conclusions

Disease-specific fears improve over the course of rehabilitation, with changes in fear of dyspnea, fear of physical activity, and fear of disease progression remaining improved at six-month follow-up. For all disease-specific fears, improvements are associated with improvements in symptom burden and quality of life, but baseline levels of disease-specific fears continue to exert their impact across rehabilitation and follow up. These findings suggest that treatments which further reduce disease-specific fears would be beneficial in individuals with COPD, and may improve the impact of pulmonary rehabilitation on health status and quality of life.

## Figures and Tables

**Figure 1 jcm-08-01460-f001:**
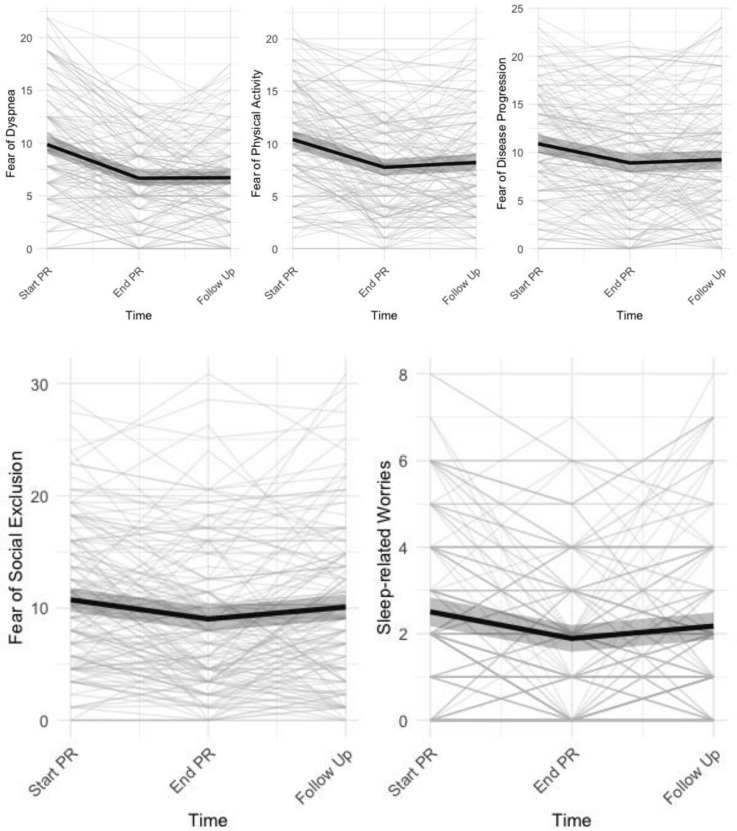
Individual trajectories of dyspnea-related fears with mean trajectories (±95% CI) superimposed.

**Figure 2 jcm-08-01460-f002:**
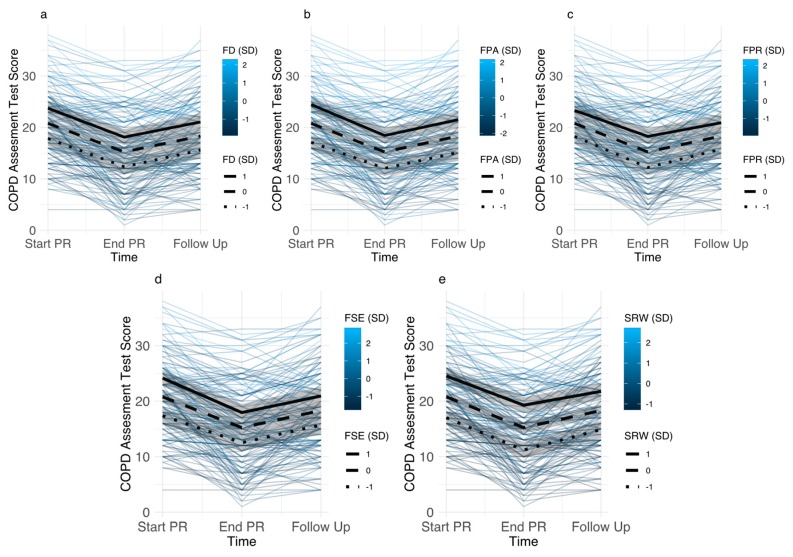
Individual trajectories of COPD assessment test (CAT) scores for participants with different levels of disease-specific fears (color indicates level of disease-specific fear). Estimated trajectories for participants with disease-specific fear scores 1SD above mean, mean, and 1SD below mean (95% CI) are superimposed. (**Panel a**): Fear of Dyspnea (FD), (**Panel b**): Fear of Physical Activity (FPA), (**Panel c**): Fear of Progression (FPR), (**Panel d**): Fear of Social Exclusion (FSE), (**Panel e**): Sleep-Related Worries (SRW).

**Figure 3 jcm-08-01460-f003:**
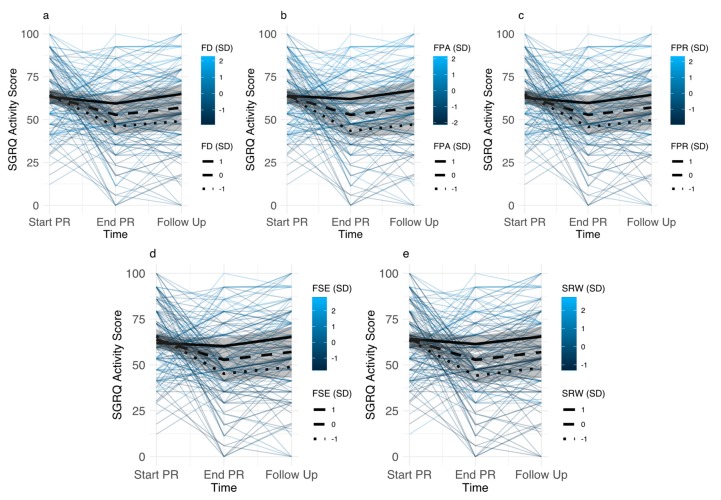
Individual trajectories of St. Georges respiratory questionnaire (SGRQ) activity scores for participants with different levels of disease-specific fears (color indicates level of disease-specific fear). Estimated trajectories for participants with disease-specific fear scores 1SD above mean, mean, and 1SD below mean (95% CI) are superimposed. (**Panel a**): Fear of Dyspnea (FD), (**Panel b**): Fear of Physical Activity (FPA), (**Panel c**): Fear of Progression (FPR), (**Panel d**): Fear of Social Exclusion (FSE), (**Panel e**): Sleep-Related Worries (SRW).

**Figure 4 jcm-08-01460-f004:**
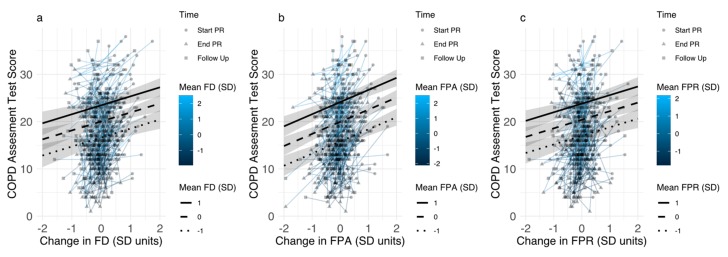
Associations between individual changes in disease-specific fears and COPD assessment test scores for participants with different levels of disease-specific fears (color indicates mean level of disease-specific fear). Estimated associations for participants with disease-specific fear scores 1SD above mean, mean, and 1SD below mean (95% CI) are superimposed. (**Panel a**): Fear of Dyspnea (FD), (**Panel b**): Fear of Physical Activity (FPA), (**Panel c**): Fear of Progression (FPR), (**Panel d**): Fear of Social Exclusion (FSE), (**Panel e**): Sleep-Related Worries (SRW).

**Table 1 jcm-08-01460-t001:** Baseline characteristics of participants.

Variable	*N*	Data	Association with Baseline Disease-Specific Fears *p <* 0.05
Age, mean (SD)	146	57.5 (4.4)	ns
Gender, *n* (%)	146		ns
Males		98 (67)	
Females		48 (33)	
Smoking status, *n* (%)	146		SRW
Active smokers		66 (45)	
Non-smokers		80 (55)	
Pack years, mean (SD)	142	44.4 (22.8)	ns
Exacerbation history (past year), *n* (%)	140		FD, FP, FSE
0 or 1 exacerbation		68 (49)	
2 or more exacerbations		72 (51)	
Inhaled Medication, *n* (%)	146		ns
ICS + LABA + LAMA		64 (44)	
ICS + LABA/LAMA		15 (10)	
LABA + LAMA		50 (34)	
LABA/LAMA monotherapy		10 (7)	
SABA or no inhaled medication		7 (5)	
FEV1% predicted, mean (SD)	142	55.4 (16.6)	FD, FPA, FSE
6MWT, mean (SD)	144	460.8 (102.5)	FD, FPA, FSE
Baseline PHQ-9, mean (SD)	138	7.9 (5.4)	FD, FPA, FP, FSE, SRW

SD = standard deviation; ICS: inhaled corticosteroid; LABA: long-acting ß-agonist; LAMA: long-acting muscarinic antagonist; SABA: short-acting ß-agonist; FEV1%: forced expiratory volume; 6MWT: Six-Minute Walking Test; PHQ-9: Patient Health Questionnaire. FD: Fear of Dyspnea, FPA: Fear of Physical Activity, FPR: Fear of Progression, FSE: Fear of social exclusion, SRW: Sleep-Related Worries. Differences in *N* are due to missing values.

**Table 2 jcm-08-01460-t002:** Change in disease-specific fears and other pulmonary rehabilitation outcomes.

		Start PR		End PR		Follow-Up				
Variable		Mean	SE	Mean	SE	Mean	SE	*F*	*df*	*p*
Disease-Specific Fears (CAF)										
	Fear of Social Exclusion	10.74 a	0.542	9.03 b	0.555	10.09 a	0.582	8.376	(2, 146.16)	0.0004
	Fear of Dyspnea	9.86 a	0.44	6.67 b	0.372	6.72 b	0.337	53.203	(2, 146)	<0.0001
	Fear of Physical Activity	10.4 a	0.425	7.75 b	0.413	8.22 b	0.414	31.614	(2, 145.98)	<0.0001
	Fear of Progression	10.90 a	0.489	8.91 b	0.488	9.25 b	0.507	14.391	(2, 146)	<0.0001
	Sleep-Related Worries	2.51 a	0.17	1.89 b	0.158	2.18 ab	0.16	10.759	(2, 146)	<0.0001
Symptom Burden (CAT)		20.8 a	0.579	15.2 b	0.603	18.3 c	0.646	70.322	(2, 145.98)	<0.0001
Quality of Life (SGRQ)										
	Symptoms	61.1 a	1.81	36.6 b	1.91	52.9 c	2.03	35.853	(2,142.28)	<0.0001
	Activity	63.8 a	1.47	52.7 b	1.62	57.0 b	2.15	16.289	(2, 144.91)	<0.0001
	Impact	41.8 a	1.37	27.7 b	1.44	31.8 c	1.81	39.083	(2, 145.69)	<0.0001
Transition Dyspnea Index (TDI)				1.92 a	0.107	1.04 b	0.135	45.484	(1, 169.9)	<0.0001
Depressive Symptoms (PHQ-9)		7.9 a	0.422	5.02 b	0.406	7.12 a	0.407	31.592	(2, 143.24)	<0.0001
6 Min walking test Distance (m)		461 a	6.68	542 b	9.24			81.711	(1, 226.17)	<0.0001

Note: Within a row, different subscripts (e.g., a, b, c) indicate means that show a significant (*p* < 0.05) difference between Start PR, End PR or Follow-up, respectively. Means with the same subscript do not show a significant difference between Start PR, End PR or Follow-up, respectively. Model for transition dyspnea index includes baseline dyspnea index values as covariate. Scale ranges are 0–32 (fear of social exclusion), 0–20 (fear of dyspnea), 0–24 (fear of physical activity, fear of disease progression), 0–8 (sleep related worries), 0–40 (CAT), 0–100 (SGRQ scales), −9–9 (TDI).

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
