# Peer review of "Impact of Disease-Specific Fears on Pulmonary Rehabilitation Trajectories in Patients with COPD"

_jcm, 2019, doi:10.3390/jcm8091460_

Round 1
Reviewer 1 Report
No further suggestions
Author Response
Thank you for reviewing this manuscript.
Reviewer 2 Report
I am satisfied with most of the authors' amendments to this manuscript, however do not understand their footnote to Table 2 regarding the subscript characters 'a,b,c'. I would recommend this be clarified for readers.
Author Response
Thank you for your review. Below, you can find a response to your question
I am satisfied with most of the authors' amendments to this manuscript, however do not understand their footnote to Table 2 regarding the subscript characters 'a,b,c'. I would recommend this be clarified for readers.--> we have further clarified this note under Table 2 by stating "Within a row, different subscripts (e.g. a,b,c) indicate means that show a significant (p<.05) difference between Start PR, End PR or Follow-up, respectively. Means with the same subscript do not show a significant difference between Start PR, End PR or Follow-up, respectively.", and hope this takes away any confusion.
This manuscript is a resubmission of an earlier submission. The following is a list of the peer review reports and author responses from that submission.
Round 1
Reviewer 1 Report
This study examined disease-specific fears in patients with COPD who underwent inpatient pulmonary rehabilitation. This is an interesting study which used multiple questionnaires to measure different aspects relating to disease-specific fears, health status, activity, quality of life and depressive symptoms.
The end of the introduction is somewhat confusing as it ends with results. This could be improved by removing the last paragraph. It would be useful to make it more clear what the primary outcome measures are. This is not stated till in the results section on page 6, line 231.
Results:
Exacerbation history would be important to include as this may have an impact on the patient's self-reported measures.
How many patients underwent psychological intervention and social counselling as part of the PR program? What effect did this have on patient fears?
What medications were subjects taking? Could their medications also had an effect of their symptoms and fears?
Table 2 - it is not clear what a,b,c refer to
A lot of data and figures are presented. The main message/results are diluted because there are so many results from many different questionnaires. Could some of this be simplified?
The colour scheme used for the figures are not particularly helpful. Could there be a better way to present the data?
Discussion:
Would be useful to discuss any findings from outpatient PR programs which are commonly used and how these results compare. Are there any studies assessing patient fears or anxiety in COPD patients participating in outpatient PR programs.
Minor comments:
Introduction line 44: "the active ingredients"
These words make this seem like this is a recipe. Suggest using different terminology.
Page 2, line 65: "proximal associations"
The word proximal in this context does not really make sense
Reviewer 2 Report
The current study was trying to investigate whether COPD-specific fear impacts the outcome of pulmonary rehabilitation. The study was well planned and conducted, data was reasonably presented. However, similar studies have been published by other groups in recent years, the novelty of the study is low, authors need to clearly state how this study is advancing over previous studies, and what's the limitations and translational importance etc.Reviewer 3 Report
The authors report upon an interesting aspect of pulmonary rehabilitation (PR) that has yet received much attention. The impact of disease-specific fears and avoidance behaviours may seriously affect the impact of PR in people with COPD and therefore warrants closer inspection. To this effect, the study is clinically significant and relatively novel.
Major comments:
· The manuscript reads very long, especially the introduction and results section. While I found the introduction to be very informative, it was far longer than necessary. I found the manuscript a bit difficult to read overall and would strongly urge the authors to prioritise succinctness over comprehensiveness of the topic in general. I feel the same message could be conveyed in a simpler/shorter way and it would make a big difference to its current readability. For example (just one example), is it crucial to report F value and p values if you are already presenting means/95%CIs for all 5 sub-domains for each result? This would reduce the heavy repetitive data burden (e.g. lines 340-367).
· Why are the outcome data that were collected at 6-weeks after PR not reported or considered in any of the analyses at all? This seems a potentially concerning omission.
· The graphs, while informative, appeared unnecessarily complicated as a reader. I’m not sure why mean data points+/-error bars weren’t just used, without the many superimposed lines, which made for very ‘busy’ images. Keep in mind that, while described as 5 figures, you are actually expecting the reader to look over 25 individual graphs, which is a lot.
· The clinical significance of the changes in the CAF have not been described. The changes may have been statistically significant, but the reader has little insight into what these magnitude of score changes represent. It is imperative that the authors articulate this to the readers.
· Related to this point above, main results (e.g. 3.2): The authors state that all subscales improved at end-of PR and that improvements were maintained at 6months. I find this difficult to defend without some clinical context (e.g. clinical significance/MID for this outcome). If I look at the data/graphs it is clear that there is a reduction in scores for all subscales from start to end of PR, but all appear to then increase (albeit slightly) after end-PR. Without clinical significance, statements such as improvements were maintained at 6 months should be reviewed and more carefully worded.
· Lines 102-111: what study is this information related to? The main one? Or the present one? Seems a very unusual place to describe this information. If this is referring to data from the current study, this should be deleted.
· Line 210-211: Why was ‘an initial sample of 146 patients’ chosen? Why were data from both arms of the study combined? How were these decisions justified? Might treatment effects from the larger study not potentially affect results?
· The analysis method used to compare patients according to the specific cutoffs (i.e. those with average scores vs those with scores 1SD above the mean) did not read explicitly clear in the methods/analysis section of the manuscript. I would make clear that you were only planning to use the one cutoff threshold of +1SD vs mean. This needs to be reviewed and amended.
· The abstract includes no data at all and is far too general – key results should be included in numerical terms.
· The study needs to be more clearly articulated as a sub-analysis of a portion of participants in the STAR study. This includes in the abstract and the start of the results section. The potential implication of only using 146 of the participants from the larger study should be acknowledged in a limitation section of the paper (which currently does not exist).
Minor issues:
· Line 114: What does the ‘first wave’ mean?
· Line 116: missing close bracket
· Line 141: were two 6MWTs conducted? Was this test performed to ATS/ERS standards? What length corridor was used? These issues need to be described in PR literature related to 6MWT.